# Ultraprocessed Food: Addictive, Toxic, and Ready for Regulation

**DOI:** 10.3390/nu12113401

**Published:** 2020-11-05

**Authors:** Robert H. Lustig

**Affiliations:** 1Department of Pediatrics, University of California, San Francisco, CA 94143, USA; Robert.Lustig@ucsf.edu; 2Institute for Health Policy Studies, University of California, San Francisco, CA 94143, USA; 3Department of Research, Touro University-California, Vallejo, CA 94592, USA

**Keywords:** processed food, nutrition, non-communicable disease, metabolic syndrome, diabetes, addiction, policy

## Abstract

Past public health crises (e.g., tobacco, alcohol, opioids, cholera, human immunodeficiency virus (HIV), lead, pollution, venereal disease, even coronavirus (COVID-19) have been met with interventions targeted both at the individual and all of society. While the healthcare community is very aware that the global pandemic of non-communicable diseases (NCDs) has its origins in our Western ultraprocessed food diet, society has been slow to initiate any interventions other than public education, which has been ineffective, in part due to food industry interference. This article provides the rationale for such public health interventions, by compiling the evidence that added sugar, and by proxy the ultraprocessed food category, meets the four criteria set by the public health community as necessary and sufficient for regulation—abuse, toxicity, ubiquity, and externalities (How does your consumption affect me?). To their credit, some countries have recently heeded this science and have instituted sugar taxation policies to help ameliorate NCDs within their borders. This article also supplies scientific counters to food industry talking points, and sample intervention strategies, in order to guide both scientists and policy makers in instituting further appropriate public health measures to quell this pandemic.

## 1. Introduction: Pandemics and Public Health

We are in the midst of two pandemics. The COVID-19 pandemic had an identifiable start in January 2020. Yet despite media attention and warnings from scientists, many countries are experiencing a “second wave”; here in the United States, we never even cleared the first wave. There is no cure, at least not yet; all we have to mitigate this pandemic are public health efforts—social distancing, handwashing, and face masks—which do not seem to work very well voluntarily, unless made mandatory by authorities. The second pandemic, of non-communicable diseases (NCDs; type 2 diabetes, cardiovascular disease, fatty liver disease, hypertension, heart disease, stroke, cancer, and dementia), has been more insidious, slowly building over a 50-year time frame [1]. There is also no cure for this pandemic; all we have are educational efforts such as voluntary “diet and exercise”, which do not seem to work very well either. 

NCDs now account for 72% of deaths [2] and 75% of health care dollars in the United States [3] and globally [2]; and the morbidity, mortality, and economic costs continue to climb. In the U.S., Medicare is expected to be insolvent by 2026, and Social Security will be broke by 2034 [4], due to both the loss of economic productivity combined with increased healthcare expenditures. Without young and healthy people paying into the system, old and infirm people cannot take out. The cost of these diseases is not limited to the U.S. [5], and NCDs have been declared a global health crisis by the United Nations (U.N.) [6]. Thus, NCDs pose an existential threat to the survival of each country, and indeed the entire planet. Identifying the cause(s) of NCDs, and upstream policy initiatives to mitigate them is of paramount importance.

Nonetheless, the world has recently faced down two other chronic disease pandemics, tobacco and ethanol; both caused by hedonic substances readily available for purchase, and both responsive to public health regulatory interventions. It was not until the U.S.’s Master Settlement Agreement and the World Health Organization (WHO) Framework Convention on Tobacco Control that we saw a reduction in cigarette consumption and reduction in lung cancer [7]. For alcohol, individual countries have passed their own public health ethanol regulatory efforts, with clear improvements [8].

## 2. Criteria for Public Health Regulation

The question for public health officials is whether there is something specific and identifiable that could be regulated on a global scale that could help to mitigate the pandemic of NCDs. While some behaviors can be mandated (e.g., mask-wearing), most are left up to each individual (e.g., exercise). Rather, targeting a substance or class of causative substances would be more effective, as predicted by the *Iron Law of Public Health*, which states that reducing availability of a substance reduces consumption, which reduces health harms [9]. Public health officials have identified the four criteria which must be met in order to be considered for public health regulation [10]: Abuse (why can’t you stop?)Toxicity (why do you get sick?)Ubiquity (why can’t you escape it?)Externalities (why does your consumption harm me?)

To generate enthusiasm for any public health regulatory effort, the science and the logic of each of these criteria must be obvious and inescapable. The goal of this treatise is to provide the science that ultraprocessed food in general, and sugar in particular, meet all four criteria, and should be considered as targets for regulation of the NCD pandemic by the public health community and by policymakers.

However, first we must deal with the “elephant in the room”; the mythology that calories are the cause of obesity, and obesity is the cause of NCDs. If this were the case, then the processed food industry can use the mantra that “any calorie can be part of a balanced diet”, and thus deflect criticisms of their products. In order to provide evidence for the specific roles of sugar and ultraprocessed food in the pandemic of NCDs, we must first confront and dispel this mythology, by demonstrating that obesity is not a cause of NCDs because normal-weight individuals get NCDs as well. We must also demonstrate that the effects of sugar and ultraprocessed food on NCD prevalence and severity are exclusive of inherent calories, and independent of effects on obesity [11].

## 3. Obesity Is a ‘Marker’, Not a Cause of Non-Communicable Diseases (NCDs)

Most clinicians mistakenly attribute the growing rise of NCDs to growing prevalence of obesity because of the *quantity of the food* ingested. This is untrue, for five separate reasons. (a) While obesity prevalence and diabetes prevalence correlate, they are not concordant [12]. There are countries that are obese without being diabetic (such as Iceland, Mongolia, and Micronesia), and there are countries that are diabetic without being obese, such as India, Pakistan, and China (they manifest a 12% diabetes rate). This is further elaborated looking at years of life lost from diabetes vs. obesity [13]. (b) Twenty percent of individuals with obesity are metabolically healthy and have normal life spans [14,15,16], while up to 40% of normal weight adults harbor metabolic perturbations similar to those in obesity, including type 2 diabetes mellitus (T2DM), dyslipidemia, non-alcoholic fatty liver disease (NAFLD), and cardiovascular disease (CVD) [17,18]. Indeed, in the U.S. 88% of adults exhibit metabolic dysfunction [19], while only 65% are overweight or obese—some normal weight people are metabolically ill as well. (c) The “Little Women of Loja” are a founder-effect cohort in Ecuador who are growth hormone-receptor deficient, and who become markedly obese yet are protected from chronic metabolic disease such as diabetes and heart disease [20]. (d) The secular trend of diabetes in the U.S. from 1988 to 2012 has demonstrated a 25% increase in prevalence in both the obese *and* the normal weight population [21]. (e) The aging process does not explain T2DM, as children as young as the first decade now manifest these same biochemical processes [22,23]. Now children get two diseases that were never seen before in this age group—T2DM and fatty liver disease. These two diseases used to be prevalent only in the elderly, or in those who abused ethanol. 

These five lines of reasoning argue that obesity is a “marker” for the pathophysiology of NCDs (e.g., insulin resistance), but not a primary cause—because a percentage of normal weight people get NCDs as well, while a percentage of people with obesity are metabolically healthy. If obesity was a cause of NCDs, then one could by extension make the case that “eating is addictive”—but clearly neither are true. That young and normal weight people can contract these diseases suggests an exposure, rather than a behavior, at the root of the NCD pandemic, and that the *quantity of the food is not the cause.*

## 4. Ultraprocessed Food Is the Cause of NCDs

Rather, *the quality of the food is the cause.* Ultraprocessed food, defined as industrial formulations typically with 5 or more ingredients [24], is the category of food that drives NCDs [25], such as obesity [26,27], diabetes [28], heart disease [29], and cancer [30]. In particular, added sugar (i.e., any fructose-containing sweetener; sucrose, high-fructose corn syrup, maple syrup, honey, agave) is the prevalent, insidious, and egregious component of ultraprocessed food that drives that risk. 

In this article, using scientific and legal evidence, I will elaborate three related arguments. First, I will demonstrate that ultraprocessed food is addictive because of the sugar that is added to it, and that the food industry specifically adds sugar because of its addictive properties. Second, I will highlight the specific mechanisms by which sugar is toxic to the liver, which leads to NCDs. Lastly, I will argue that added sugar is more appropriately defined as a *food additive* rather than as a *food.* In so doing, I will argue that added sugar, and by extension the entire ultraprocessed food category, meets these criteria established by the public health community for regulation of a substance (abuse, toxicity, ubiquity, externalities) [9]. 

## 5. Added Sugar Is Abused

The seminal role of the Western Diet in the pandemic of NCDs is unchallenged [31]. For instance, ultraprocessed food consumption correlates with body mass index (BMI) in the U.S. [26] and in 19 European countries [27]. As market deregulation policies of the 1990s took hold, fast food sales increased incrementally in all countries and cultures to which it has been introduced, along with commensurate increases in BMI [32]. Indeed, every country that has adopted the Western diet is burdened with the development of NCDs and their resultant costs [33]. However, the food industry continues to promulgate the argument that it is the quantity, not the quality of the foods that are to blame. This is not a semantic argument. Quantity is determined by the end user, a personal responsibility issue; while quality is determined by manufacturers, a public health issue. But what if the quality altered the quantity? Those that favored either view over the other would thus appear to be justified within their own stance. Indeed, this debate seems to have drawn to an academic stalemate [34,35,36]. This must be answered before any form of societal intervention can be contemplated.

### 5.1. ‘Food Addiction’ versus ‘Eating Addiction’

Recent revelations in the popular literature have alluded to the addictiveness of the Western diet [37,38], driving excessive consumption. Physiologic [39,40] and neuroanatomic [41] overlap between obesity and addiction pathways have been elucidated. Some investigators have argued that specific components of processed food, and in particular those in “fast food”, are addictive in a manner similar to cocaine and heroin [42,43]. The Yale Food Addiction Scale (YFAS) logs specific foods as having addictive properties [44], and a children’s YFAS also reveals that food addiction is common, especially in obese youth [45].

However, not everyone subscribes to this expanded view of specific foods having addicting properties. For instance, a group of academics in Europe called NeuroFAST does not accept the concept of food addiction, rather calling it “eating addiction” [46]. This group has proffered its own “eating addiction scale” in which all foods are treated similarly [47], and it is the behavior that distinguishes the phenomenon. These investigators state that even though specific foods can generate a reward signal, they cannot be addicting because they are essential to survival. In their own words:

“In humans, there is no evidence that a specific food, food ingredient or food additive causes a substance-based type of addiction (the only currently known exception is caffeine which via specific mechanisms can potentially be addictive). Within this context we specifically point out that we do not consider alcoholic beverages as food, despite the fact that one gram of ethanol has an energy density of 7 kcal [48]”.

NeuroFAST recognizes caffeine as addictive, but gives it a pass. Xanthine alkaloids are present naturally in many foods, yet caffeine is classified by the U.S. Food and Drug Administration (FDA) as a food additive. It is also a drug; we give it to premature newborns with underdeveloped nervous systems to stimulate the central nervous system (CNS) to prevent apnea. NeuroFAST also recognizes ethanol as addictive, and also gives it a pass. Natural yeasts constantly ferment fruit while still on the vine or tree, causing it to ripen [49], yet NeuroFAST acknowledges that purified ethanol is not a food. Rather, ethanol is a drug; we used to give it to pregnant women to stop premature labor.

Recently, another European group with food industry ties assessed the effects of specific foodstuffs on “eating dependence” in a cohort of university students, using weight gain as the metric of food addiction. In their study, they found no difference between fats and sugars as cause for weight gain [50]. However, as stated earlier, using weight gain as the metric of food addiction is inherently flawed.

In order to assess mechanism of effects of food on the addiction pathway in the brain, our group at UCSF studied a cohort of postmenopausal women with obesity who received orally the mu-opioid receptor antagonist naltrexone as a probe of the brain’s reward system. We found that the amplitude of cortisol responses and nausea generation in response to naltrexone correlated with symptoms of craving for sweet palatable foods in these women. These data suggest that naltrexone interfered with endogenous opioid peptide (EOP) tone that mediated these cravings. In so doing, we have discerned a phenomenon of “Reward Eating Drive” (RED), which belies those individuals with obesity who appear to respond excessively to hedonic food cues [51,52,53], and which is tied to the opioidergic component of the reward system in the brain, which is driven by sweet foods. Furthermore, using functional magnetic resonance imaging (fMRI) studies, other investigators have defined the prefrontal cortex as responsible for the response of sweet tastes as being “attractive” or “unattractive” [54].

### 5.2. Addictive Potential of Food Components

If there was a class of consumables that was uniquely addictive, it would have to be “fast food”. But is it just the calories, or is there something specific about fast food that generates an addictive response? Fast food contains four components whose hedonic properties have been examined: salt, fat, caffeine, and sugar [37,42].

#### 5.2.1. Salt

In humans, salt intake has traditionally been conceived as a learned preference [55] rather than as an addiction. The preference for salty foods is likely learned early in life. Four- to six-month-old infants establish a salt preference based on the sodium content of breast milk, water used to mix formula, and diet [56]. Because energy-dense fast foods are relatively high in salt [57], in part as a preservative to reduce depreciation, the preference for salty foods is associated with higher calorie intake. For example, a study in Korean teens showed a correlation between frequent fast food intake and preference for saltier versions of traditional foods [58]. Another study examined 27 subjects undergoing opiate (mostly oxycodone) withdrawal and showed significant increases in fast food intake and weight gain over 60 days [59], suggesting “addiction transfer”. On the other hand, studies show that people can ‘reset’ their preference for less salty items. This has been demonstrated in adolescents deprived of salty pizza on their school lunch menu, and hypertensive adults who were retrained to consume a lower sodium diet over 8 to 12 weeks [55]. Furthermore, at low levels, salt intake is well known to be tightly regulated. For example, patients with salt-losing congenital adrenal hyperplasia who lack the mineralocorticoid aldosterone modulate have an obligatory salt loss, which modulates their salt intake [60], until appropriate doses of fludrocortisone are supplemented. The notion that human sodium intake is “physiologically fixed” had been used to criticize recent public health efforts to reduce sodium intake so drastically [61]. Nonetheless, the U.K. government engaged in a secret mass campaign to reduce public salt consumption by 30%, and saw a 40% reduction in hypertension and stroke without signs of withdrawal [62]. 

#### 5.2.2. Fat

The high fat content of fast food is vital to its rewarding properties. Indeed, there may be a “high-fat phenotype” among human subjects, characterized by a preference for high-fat foods and weak satiety in response to them, which acts as a risk factor for obesity [63]. However, so-called “high-fat foods” preferred by people are almost always also high in carbohydrate (e.g., potato chips, pizza, or cookies). Indeed, adding sugar significantly enhances preference for high-fat foods among normal weight human subjects; yet there was no limit for preference with increasing fat content [64]. Thus, the synergy of high fat along with high sugar is likely to be more effective at stimulating addictive overeating than fat alone. However, these rewarding properties of fat appear to be strictly dependent on simultaneous ingestion of carbohydrate, as low-carbohydrate high-fat (LCHF) [65] and ketogenic diets [66] consistently result in reduced caloric intake, significant weight loss, and resolution of metabolic syndrome. In other words, fat increases the salience of fast food, but does not appear to be addictive in and of itself.

#### 5.2.3. Caffeine

Caffeine is a “model drug” of dependence in humans [67], meeting the DSM-IV and DSM-5 criteria for tolerance, physiologic withdrawal, and psychological dependence in children [68], adolescents [69], and adults [70]. Headache [70], fatigue, and impaired task performance [68] have been demonstrated during withdrawal. While adolescents and children get their caffeine from soft drinks and hot chocolate, adults get most of their caffeine from coffee and tea [71]. These drinks average 239 calories and provide high amounts of sugar [72]. Soft drink manufacturers identify caffeine as a flavoring agent in their beverages, but only 8% of frequent soda drinkers can detect the difference in a blinded comparison of a caffeine-containing and caffeine-free cola [73]. Thus, the most likely function of the caffeine in soda is to increase the salience of an already highly rewarding (high sugar) beverage. These drinks may be acting as a gateway for caffeine-dependent customers to visit a fast food restaurant and purchase fast food [74].

#### 5.2.4. Sugar

Other than caffeine, the component with the highest score on the YFAS is sugar [44]. Adding a soft drink to a fast food meal increases the sugar content 10-fold. Multivariate analysis of fast food transactions demonstrate that only soft drink intake is correlated with changes in BMI; not animal fat products [32]. While soda intake has been shown to be independently related to obesity and the diseases of metabolic syndrome [75,76], fast food eaters clearly consume more soft drinks. Sugar has been used for its analgesic effect in neonatal circumcision [77], suggesting a link between sugar and EOP tone. Indeed, anecdotal reports from self-identified food addicts describe sugar withdrawal as feeling “irritable”, “shaky”, “anxious” and “depressed” [78]; symptoms also seen in opiate withdrawal. Other studies demonstrate the use of sugar to treat psychological dependence [79]. Sugar craving can vary widely by age, menstrual cycle and time of day [80].

Sugar is added to food either as sucrose, high-fructose corn syrup (HFCS), honey, maple syrup, or agave. In general, each are assumed to consist of half fructose, half glucose; although this percentage has recently come into question when an analysis of store-bought sodas in Los Angeles revealed a fructose content as high as 65% [81]. This difference may be relevant, as fructose appears to generate a greater reward response and more toxicity than does glucose (see below).

### 5.3. Correlates of Addiction in Animals Exposed to Sucrose

In rodents, oral sucrose administration uniquely induces the acute reactant *c-fos* in the ventral tegmental area, implying activation of the reward pathway [82]. Furthermore, sucrose infusion directly into the nucleus accumbens reduces dopamine and µ-opioid receptors similar to morphine [83], and fMRI studies demonstrate the establishment of hard-wired pathways for craving [84]. Furthermore, sucrose administration to rodents induces behavioral alterations consistent with dependence; i.e., bingeing, withdrawal, craving, and cross-sensitization to other drugs of abuse [85]. Indeed, in one oft-quoted rat study, sweetness surpassed cocaine as reward [86].

### 5.4. Differential Effects of Fructose vs. Glucose vs. Fat on the Human Brain

Despite being calorically equivalent (4.1 kcal/gm), fructose and glucose are metabolized differently. Glucose is the energy of life. Glucose is so important that if you do not consume it, your liver makes it from amino acids and fatty acids (gluconeogenesis). Conversely fructose, while an energy source, is otherwise vestigial; there is no biochemical reaction in any eukaryote that requires it. Our research has shown that when provided in excess of the liver’s capacity to metabolize fructose via the tricarboxylic acid cycle, the rest is turned into liver fat, promoting insulin resistance, and resultant NCDs [87,88,89].

Physiologically, chronic fructose administration promotes fasting hyperinsulinemia and hypertriglyceridemia [90], which blocks leptin’s ability to cross the blood brain barrier [91], and attenuates leptin’s ability to extinguish mesolimbic dopamine signaling in rodents [92] and humans [93], thus promoting tolerance and withdrawal [94]. Furthermore, fructose does not suppress the stomach-derived hunger hormone ghrelin [95]. Through these pathways, fructose fosters overconsumption independent of energy need [96]. A comparison of the two monosaccharides demonstrates increased risk for bingeing with fructose (similar to sucrose) as opposed to glucose [97], suggesting the fructose molecule is the moiety that generates both reward and addiction responses.

Neuroanatomically, human fMRI studies show that acute glucose vs. fructose administration exert effects on different sites in the brain. One study infused each monosaccharide intravenously, and measured blood oxygenation level-dependent (BOLD) fMRI signal in cortical areas of the brain; glucose increased the BOLD signal in cortical executive control areas, whereas fructose suppressed the signal coming from those same areas [98]. Another study examined regional cerebral blood flow (rCBF) after oral glucose vs. fructose. With glucose, rCBF within the hypothalamus, thalamus, insula, anterior cingulate, and striatum (appetite and reward regions) was reduced, while fructose reduced rCBF in the thalamus, hippocampus, posterior cingulate cortex, fusiform, and visual cortex [99]. Consistent with other studies, fructose demonstrated lack of satiety or fullness in comparison to glucose. Furthermore, glucose increased “functional connectivity” of the caudate, putamen, precuneus, and lingual gyrus (basal ganglia) more than fructose; whereas fructose increased functional connectivity of the amygdala, hippocampus, parahippocampus, orbitofrontal cortex and precentral gyrus (limbic system) more than glucose [100]. In obese youth, the effects of oral fructose on dopamine activation of the nucleus accumbens is severely attenuated, suggesting down-regulation of dopamine receptors [101]. Lastly, the effects of fat and sugar both separately and together (adjusting for calories) on fMRI signaling have been assessed [102]. High-fat milkshakes increased brain activity in the caudate and oral somatosensory areas (postcentral gyrus, hippocampus, inferior frontal gyrus); while sugar increased activity in the insula extending into the putamen, the Rolandic operculum, and thalamus (gustatory regions). Furthermore, increasing sugar caused greater activity in those regions, but increasing fat content did not alter this activation. In other words, the fat increases the salience of the sugar, but it is the sugar that effectively recruits reward and gustatory circuits.

To summarize, added sugar (and specifically the fructose moiety) is unique in activating reward circuitry; fructose works both directly and indirectly to increase consumption; and that both obesity and chronic fructose exposure down-regulate dopamine receptors, requiring greater and greater stimuli to enact a reward-signaling effect (tolerance), a primary component of addiction.

### 5.5. ‘Food’ Addiction Is Really ‘Food Additive’ Addiction, and ‘Added Sugar’ Is a Food Additive

In the past, the concept of food addiction was not embraced by psychiatrists. For instance, the DSM-IV published in 1993 listed “substance use disorder” as requiring both tolerance and withdrawal as necessary criteria for the definition of addiction, and (apart from caffeine and ethanol) no foodstuff elicited withdrawal. However, as the public health difficulties stemming from addiction expanded, the definition, of necessity, expanded. The DSM-5 published in 2013 reclassified the field so as to include “behavioral addictions”, such as gambling (internet gaming was included in the Appendix as “requiring further study”). Thus, a revised set of criteria related to psychological dependence was proffered [103], including:Craving or a strong desire to use;Recurrent use resulting in a failure to fulfill major role obligations (work, school, home);Recurrent use in physically hazardous situations (e.g., driving);Use despite social or interpersonal problems caused or exacerbated by use;Taking the substance or engaging in the behavior in larger amounts or over a longer period than intended;Attempts to quit or cut down;Time spent seeking or recovering from use;Interference with life activities;Use despite negative consequences.

However, food addiction was not codified in the DSM-5. Nonetheless, systematic reviews of the literature demonstrate that ultraprocessed foods have the highest addictive potential due to their added sugar content [104]. While sugar itself does not exhibit the DSM-IV criteria of classic tolerance and withdrawal, sugar clearly meets the DSM-5 requirements of tolerance and dependence (use despite conscious knowledge and recognition of their detriment).

Coca leaves are medicinal in Bolivia, yet cocaine is a drug, and regulated. Opium poppies are also medicinal, but morphine is a drug, and regulated. Caffeine is found in coffee (medicinal for many), yet concentrated caffeine is a drug, and regulated. In ancient times, sugar was a spice. Through the Industrial Revolution it was a condiment. Now it is purified, and it is a drug. Refined sucrose is the same compound found in fruit, but the fiber has been removed, and it has been crystallized for purity. This process of purification has turned sugar from “food” into “drug” [105]. Like these other addictive consumables, it can be present in low dose in nature and not exert toxic effects; but when purified and added to food, it becomes addictive.

Drugs are a luxury, food is a necessity. NeuroFAST asks how can foods that are necessary to survival also be addicting? Because certain “foods” are *not* necessary for survival. Of the hedonic substances found in food, only alcohol, caffeine, and sugar are addictive. But these are food additives, not foods. Some form of sugar has been added to 74% of the food supply [106], because the food industry knows that when they add it, we buy more [107]. For instance, the tobacco industry manipulated nicotine levels in cigarettes specifically to keep users consuming, and to convert as many as possible into “heavy users” [108]. The food industry has engaged in similar practices, which has increased the percent of calories as added sugar (58%) in ultraprocessed foods [27]. In fact, sugar’s allure is a big reason why the processed food industry’s current profit margin is 5% (it used to be 1%) [109]. The addictive nature of sugar is also revealed in its economics. For instance, coffee is price-inelastic, i.e., increasing price does not reduce consumption much. When prices jumped in 2014 due to decreased supply, Starbuck’s sales remained constant, owing to its hedonic effects [110]. As consumables go, soft drinks are the second most price-inelastic, just below fast food [107]. When the price is raised by 10% (e.g., with taxes), consumption dropped only 7.6%, mostly among the poor, as was seen in Mexico [111]. Thus, sugar consumption is only minimally responsive to either its economic or caloric value, consistent with its addictive properties.

## 6. Added Sugar Is Toxic

Toxicity is defined as “the degree to which a substance can damage an organism”. Such detrimental effects must be exclusive of caloric equivalence, or else *all* calories are toxic, which is clearly not true. Just because a substance is an energy source does not mean that it is not toxic. For instance, alcohol possesses a caloric equivalence (7 kcal/gm), yet we humans have an upper limit of hepatic and brain metabolism, beyond which toxicity becomes manifest, either acute (mental status changes) or chronic (fatty liver disease progressing to cirrhosis, insulin resistance). Alcohol is not dangerous because of its calories or its effects on weight. Alcohol is dangerous because it is alcohol [112]; the biochemistry of the molecule in the liver confers its toxicity. Alcohol exerts its negative effects on liver metabolism through two mechanisms: (1) liver mitochondrial overload with diversion of substrate to the process of *de novo* lipogenesis (DNL; new fat-making), with subsequent hepatic fat accumulation and insulin resistance [113]; and (2) the non-enzymatic binding of the intermediate metabolite acetaldehyde to liver proteins, known as the Maillard or “aging” reaction, with subsequent “carbonyl” stress (see Section 6.1.2), protein denaturation, subsequent inflammation, and cell death.

### 6.1. Detrimental Effects of Fructose on Liver Metabolism

The metabolic perturbations associated with fructose consumption exclusive of its caloric equivalence are well documented by numerous investigators [114,115]. There are no biochemical reactions that require dietary fructose. The same two primary molecular mechanisms of alcohol delineate the toxicity of fructose apart from its caloric equivalence [105].

#### 6.1.1. *De Novo* Lipogenesis

Only the liver metabolizes fructose for energy, and a fructose bolus (e.g., a soft drink) absorbed across the intestinal lining delivers the majority of the fructose via the portal vein to the liver. Fructose is particularly lipogenic, as the glycolytic intermediate acetyl-CoA is delivered to the liver mitochondria in an unregulated fashion, driving hepatic DNL, which will either be exported as triglyceride (which contributes to heart disease); or possibly overwhelming the liver’s lipid export capacity, leading to intrahepatic lipid deposition and hepatic steatosis, resulting in liver insulin resistance, which is a driving force behind all the NCDs [89]. The intermediate metabolic pathways have been elucidated elsewhere [116].

#### 6.1.2. Carbonyl Stress—The Maillard Reaction

Carbonyl stress occurs when the reactive aldehyde or keto-group of a carbohydrate molecule binds non-enzymatically to the amino-group of a protein, leading to the Maillard or the “browning reaction” [117]. This is why bananas brown as they age. This is also why humans get wrinkles as they age. This is also why patients with diabetes check their hemoglobin A1c measurement (which is a carbohydrate molecule bound to position 1 of the globin chain), to determine if their diabetes is out of control. Every time this reaction occurs, the protein becomes less flexible (leading to cell dysfunction), and an oxygen radical is produced, which if not quenched by an antioxidant, can lead to protein or lipid peroxidation, cell damage, and death.

Due to its unique stereochemistry, the ring form of fructose (a five-membered furan with axial hydroxymethyl groups) is under a great deal of ionic strain, which favors the linear form of the molecule, exposing the reactive 2-keto position, which engages in the fructosylation of exposed amino-moieties of proteins via the Maillard reaction, and 7 times faster than the 1-aldehyde position of glucose reacts with those same proteins. Each Maillard reaction generates one oxygen radical, which must be quenched by an antioxidant, or else cellular damage can ensue. Thus, due to its chemical makeup, fructose leads to increased cellular damage [118] and disease progression compared to glucose, and unrelated to its caloric equivalence.

#### 6.1.3. Tying Two Pathophysiologic Mechanisms Together—Methylglyoxal

Recently, our UCSF/Touro research group has determined that methylglyoxal, a specific intermediate in the glycolytic pathway, is likely the nidus of both of these toxic phenomena within the liver [119]. Methylglyoxal is a transient metabolic intermediate of the process of anaerobic glycolysis, whose production is dependent on the availability of excess substrate (either glucose or fructose) in the liver; but, because virtually 100% fructose load is handled by the liver, compared to only 20% of glucose, then fructose is the primary driver of its formation. Methylglyoxal is an alpha-dicarbonyl; it is both a reactive aldehyde (like glucose) and a reactive ketone (like fructose) at the same time. Therefore, it engages in the Maillard reaction 35 times faster than fructose, and 250 times faster than glucose, generating 250 times the oxygen radicals. Methylglyoxal is detoxified to the byproduct D-lactate, which can be measured in the blood, and serves as a proxy of the rate of methylglyoxal formation. D-lactate levels are higher in obese adolescents [120], and reductions in D-lactate levels by fructose restriction in obese children correlate with improvements in DNL, liver fat content, and insulin sensitivity [121], all unrelated to caloric equivalence or obesity. These findings argue that fructose is a chronic, dose-dependent hepatotoxin, which drives progression of NCDs.

### 6.2. Dissociating Added Sugar from Its Calories and Effects on Weight

The food industry often tries to divert the public health conversation toward obesity [50,122]. Sugar ranks below potato chips and French fries as a cause of weight gain [123]; the data correlating sugar consumption to obesity are weak, accounting for only about 10% of the observed effect [124]. If sugar is only one of many causes of weight gain, it can iterate its mantra, ‘a calorie is a calorie’. However, a new study demonstrates that the correlation between added sugar consumption and population obesity obeys a slightly more complex function, taking into account both current and previous consumption of added sugar [125]. This model predicts the effects of added sugar on obesity quite accurately.

But, as stated before, obesity is the wrong metric. Obesity and diabetes are discordant; there are countries where diabetes rates are high yet obesity rates are low, such as India, Pakistan, and China; while their sugar consumption has increased by 15% in the past 6 years alone [126]. When weight and calories are factored out, the correlation between sugar consumption and type 2 diabetes is even stronger [12,127]. To date, the food industry refuses to engage in a discussion on the role of added sugar in chronic metabolic diseases, exclusive of obesity.

There are many case-control studies (reviewed in [128,129]) which point to dietary fructose consumption as a primary cause of T2DM, but such studies are not controlled for calories or weight. In order to prove that fructose (and, therefore, added sugar) is specifically toxic, the molecule must be dissociated from its inherent calories and its effects on weight. Furthermore, standard cross-sectional or correlational studies without a time-factor analysis component are not acceptable, as they cannot distinguish reverse or intermediate causality; it is like the snapshot rather than the movie. Lastly, the food industry is quick to point out that most fructose studies are done in rodents, with large doses over a short period of time. In defense, a recent study in rats shows that sugar at normal levels of consumption can cause morbidity and mortality [130], and a primate study demonstrates similar detrimental effects [131]. Nonetheless, in order to prove toxicity, this section will be limited to human studies using doses of added sugar routinely consumed.

#### 6.2.1. Prospective Association Studies

Three recent studies, all controlled for calories and adiposity and with a time analysis, support sugar as a specific and direct causative agent in T2DM. First, a prospective cohort analysis of the European EPIC-Interact study found that sugar-sweetened beverage (SSB) consumption increased risk for development of diabetes over a 10-year period. The multivariate modeling, which adjusted for both energy intake (EI) and for adiposity (BMI), demonstrated that each SSB consumed increased the hazard risk (HR) ratio by 1.29 (95% CI 1.02, 1.63) exclusive of energy intake (calories) or BMI (obesity) [132]. In the U.S., we are currently consuming the equivalent of 2.5 servings of SSB’s per day; so our HR ratio is 1.68.

Second, a meta-analysis of studies isolated consumption of soda (*n* = 17) and fruit juice (*n* = 13) separately, while controlling for calories and adjusting for adiposity [76]. This meta-analysis showed that both soda and fruit juice significantly increased the relative risk (RR) ratio for diabetes (1.27, 1.10, respectively) over time. Furthermore, this study specifically took into account the fact that food industry-sponsored studies frequently demonstrate publication and information bias, and calibrated for these biases.

Third, our UCSF group evaluated the National Health and Nutrition Examination Survey (NHANES) adolescent database across three cycles 2005–2012, to determine nutritional consumption and any changes within the American diet within that interval. We then binned subjects into quintiles based on added sugar consumption, and after controlling for caloric intake and BMI, determined what aspects of the diet predicted the prevalence of metabolic syndrome [133]. We set the HR ratio for metabolic syndrome in the 1st quintile (median added sugar consumption = 30 gm/day) at 1.0; by the 4th quintile (median added sugar consumption = 125 gm/day), the HR ratio for metabolic syndrome had increased to 9.7.

#### 6.2.2. Econometric Analyses

One econometric analysis [134] of 156 countries over the period 1995–2014 demonstrated that global availability of sugar and sweeteners was correlated with diabetes prevalence, health care costs per diabetic, and health care costs per capita; demonstrating both personal and societal harm related to added sugar consumption. This analysis also showed that this correlation occurred in both developed and developing countries. However, this study did not account for calories or obesity, and could not account for other aspects of the diet.

Our UCSF/Stanford group performed an econometric analysis to assess what foods were specifically implicated in altered diabetes rates over time [12]. We melded three freely available databases together; (1) the Food and Agriculture Organization statistics database (FAOSTAT; a branch of the World Health Organization), which lists by food availability per person by country, by year 2000–2010, and by line item (total calories, fruits excluding wine, meats, oils, cereals, fiber-containing foods, and sugar/sweeteners); (2) the International Diabetes Federation (IDF) database which lists diabetes prevalence by country by year 2000–2010; and (3) the World Bank World Development Indicators Database for the years 2000–2010, in which Gross Domestic Product is expressed in purchasing power parity in 2005 US dollars for comparability among countries to control for poverty. It also controls for urbanization, aging, physical activity, and obesity. We asked what food(s) availability predict change in diabetes prevalence country by country over the decade? We performed this analysis using generalized estimating equations with a conservative fixed-effects approach (Hausman test), a hazard model to control for selection bias (Heckman selection test), and period effects controlled for secular trends that may have occurred as a result of changes diabetes detection capacity or importation policies. Most importantly, we examined longitudinal data between 2000 and 2010, which allowed us to determine what dietary changes preceded the changes in diabetes prevalence (Granger causality test).

We demonstrated that retrospective changes in sugar availability predicted the prevalence of diabetes during the decade 2000–2010, exclusive of total calories, other foodstuffs, aging, obesity, physical activity, or income. For every 150 calories per day in excess, diabetes prevalence increased 0.1%, but if those 150 calories happened to be a can of soda, diabetes prevalence increased 11-fold, by 1.1% [12]. These data meet the Bradford Hill criteria for “causal medical inference”, because we demonstrate dose (more sugar, more diabetes), duration (longer sugar exposure, more diabetes), directionality (the few countries where sugar availability went down experienced a reduction in diabetes), and precedence (we noted a three-year lag between increase in sugar availability and increase in diabetes prevalence; in a prospective modeling study we noticed a three-year lag between sugar reduction and decrease in diabetes prevalence [3]).

This econometric analysis has been criticized for two reasons. First, it is an “ecological study”, which by convention is hierarchically considered of low quality. Rather, this econometric analysis is more rigorous and of higher quality than all studies except randomized controlled trials [135], as it assesses multiple points in time, discerns complex relationships between internal and external motivating factors (adjusted over time), and allows for determination of causation (Granger causality test). Second, the FAOSTAT database assesses country-specific food availability rather than consumption, and waste is not taken into account. Rather, assessing availability is a positive feature rather than a negative, as availability is more accurate, easily quantifiable, not subject to the vicissitudes of individual recall, and independent of food wastage.

#### 6.2.3. Interventional Starch-for-Sugar Exchange

Our UCSF/Touro research group documented the effects of isocaloric substitution of sugar with starch in 43 Latino and African-American children with metabolic syndrome over a 10-day period [87,88,89]. We performed food questionnaires and interviews using sophisticated software to assess their total caloric consumption, as well as specific macronutrient and fiber intake. On Day 0, we assessed their metabolic health on their home diet using: (1) baseline analyte levels; (2) oral glucose tolerance testing; and (3) dual-emission X-ray absorptiometry (DEXA) scanning. Then, for the next 9 days, we catered their meals, to provide the same caloric content, the same fat, protein, and fiber content, and the same amount of total carbohydrate; but we reduced the percent calories from dietary sugar from a mean of 28% to 10%, and the percent calories from fructose from 12% to 4%. They were allowed fruit, but not fruit juice. We gave them a scale to take home and called them every day. If their weight was declining, we made them eat more, and they were given extra snacks to prevent weight loss. Then we studied them again 10 days later.

Every aspect of their metabolic health improved, with essentially no change in weight. Blood pressure reduced by 5 mmHg, triglycerides by 33 mg/dL, low-density lipoproteins (LDL) by 10 mg/dL, glucose levels reduced by 5 mg/dL, glucose area under the curve dropped by 8%, fasting insulin dropped by 10 mU/L, insulin area under the curve dropped 25%, on the same number of calories and without weight loss, just by removing the added sugar and substituting with starch. Furthermore, subcutaneous fat did not change (as there was no weight loss), but visceral fat reduced by 7%, and most importantly liver fat was reduced by 22%. We also showed that insulin dynamics improved markedly, thus reversing their predisposition to T2DM.

Taken together with the aforementioned studies [12,136,137], Koch’s Postulates for causation of NCDs by added sugar are fulfilled. Sugar is a chronic, dose-dependent liver toxin unrelated to calories or obesity, similar to ethanol, because fructose and ethanol exert similar effects on the liver and the brain [112].

## 7. Added Sugar Is Ubiquitous

Sugar has become ubiquitous in the Western diet, increasing from 15 gm/day at the beginning of the 20th century to 94 gm/day at the beginning of the 21st century [138,139]. In the U.S. 56% of the diet is now ultra-processed food, 62% of sugar in the American diet is in this category [28], and some form of sugar has been added to 74% of the items in the American grocery store [106], because the food industry knows that when they add it, we buy more. Similarly, world sugar consumption tripled 1960–2010 while the world population doubled over the same time [140], arguing that most of the world’s population has experienced a significant increase in added sugar consumption in the 50 years that NCDs have become prominent [140]. For instance, changes in consumption of Coca-Cola over the interval 1993–2006 correlated with changes in diabetes prevalence in both China and Mexico. During this interval, the consumer price index for sugared beverages increased 50% vs. food, and 25% vs. fruits and vegetables [141]. The introduction of high-fructose corn syrup in 1975 reduced cost the cost of sugar by 50%, which allowed serving size to rise, and sugar to be added to foods that previously did not contain it. For instance, 50% of milk sales in elementary and middle schools are for flavored milk (chocolate, strawberry). Furthermore, in most developing nations, soda is cheaper than water, which has increased consumption of added sugar around the world. Processed foods and sugared beverages are marketed heavily as they are extremely profitable; in 2006, food marketers spent USD $1.05 billion on marketing to children and adolescents; half of which were for sugared beverages [142].

Marketing practices by tobacco and food companies are highly congruent [143]. Big Tobacco in the past, and Big Food currently, have used “commercial speech” provided by the First Amendment to cull favor with the public through advertising and sponsorships. For instance, both have in the past engaged in vigorous advertising campaigns to recruit new users that was defused only by regulatory agency action [144,145]. For decades Big Tobacco provided corporate sponsorship of various public events around the world, such as the Olympics, baseball and football games, and sporting events around the world. The fast food and beverage industries engage in similar marketing practices, sponsoring global events around the world. Big Tobacco shamelessly marketed their products to children (e.g., Joe Camel); while the food and beverage industries have followed suit (e.g., Ronald McDonald). Both have used deceptive business practices to maintain increased use of their product among “heavy users” [146,147].

## 8. Added Sugar Exerts Externalities

Substances that produce societal harms impact even the non-user. Second-hand smoke and drinking-driving provided strong arguments for tobacco and alcohol control, respectively. The above data demonstrate that the long-term healthcare, human, and economic costs of NCDs place the chronic effects of fructose overconsumption in the same category [148].

Sugared beverages alone kill 184,000 people per year globally [149]. The U.S. wastes $65 billion in work productivity and $150 billion in health care resources, and experiences a 50% increase in absenteeism and health insurance premiums, all to care for the co-morbidities of metabolic syndrome [150]. Currently, 75% of all health care dollars are spent on treating these diseases or resultant disabilities. Rising global NCD rates yield an annual mortality of 35 million people, with a disproportionate 80% of these deaths occurring in low- and middle-income countries, wasting precious medical resources [151]. Lastly, the past three Surgeons General and the Chairman of the Joint Chiefs of Staff have declared obesity a “threat to national security”. The original Pentagon report from 2012 has been updated in 2018, and 33% of recruits are now deemed “Still Too Fat To Fight” [152]. Even among those recruited, 43% cannot be deployed into the field due to Stage 3 dental caries due to sugar consumption [153].

Population-wide sugar reduction would prevent premature death, save economies billions and improve quality of life for millions across the globe. Our UCSF group used advanced Markov modeling (using fatty liver disease as the sentinel disease) to demonstrate that reduction of added sugar consumption of just 20% (e.g., a tax) could reduce obesity, type 2 diabetes, heart disease, death rates, and medical expenditures within three years in the United States, and save $10 billion annually, while a 50% reduction (e.g., adhering to U.S. Dept. of Agriculture (USDA) guidelines) could save $31.8 billion annually [3]. On the productivity side, Morgan Stanley modeled global economic growth rates from 2015 to 2035 in low-sugar and high-sugar simulations [154], and showed that using a low-sugar case, economic growth would be maintained at 2.9%, while using a high-sugar case (e.g., the present), economic growth would slowly decline to 0.0%. Thus, the externalities of added sugar consumption are direct and affect everyone.

## 9. Food Industry Counters

### 9.1. Personal Responsibility

Education of the public through emphasis on “personal responsibility” over the last 30 years have not been effective in stemming the tide of obesity and metabolic syndrome. This should not be surprising, as educational efforts have been unsuccessful in reducing the consumption of other substances of abuse [9,155]. Add to this the fact that 74% of the items in the food supply are spiked with added sugar by the food industry [106]; thus it is virtually impossible for most individuals to disabuse sugar, and to be able to go “cold turkey” in order to reduce toxicity and dependence. This is especially true of the poor, who have limited access to healthy food, and are often limited in their purchases to high-sugar processed food on the Supplemental Nutrition Assistance Program (aka Food Stamps). The ostensible reason that the food industry has added more and more sugar to processed food is for “palatability”. Indeed, when they do, we buy more; which reinforces the practice by increasing profits. Indeed, efforts to reduce the negative health impact of “junk food” by former Pepsi CEO Indra Nooyi by introducing a “good for you” category (to offset their “fun for you” category) have met with rancor by her own Board of Directors due to a $349 million reduction in profits [156].

The personal responsibility strategy was first deployed by tobacco companies in 1962 as a reason to keep on smoking [157]. This ideology requires four pre-requisites:

#### 9.1.1. Knowledge

Information labelling is not easily understandable by the regular consumer buying food products in the supermarket. Many will trust and buy a product on the way it is promoted, rather than on its nutritional value. Until recently, the US Institute of Medicine, and in the UK and the rest of Europe for the past 15 years, guideline daily amounts on labels have suggested that daily consumption of up to 22 teaspoons of sugar is healthful [158].

#### 9.1.2. Access

Over 70% of foods in the supermarket contain added sugar—it has become almost unavoidable. Processed sugary food and drinks have permeated workplaces, gyms, and schools. Several American hospitals (including UCSF), and the British National Health Service (NHS) have instituted a ban on sugary drinks sold in hospitals, in order to provide a role model for the public. Our UCSF group has documented the metabolic health benefits of a workplace ban on sugared beverages [159].

#### 9.1.3. Affordability

One should be able to afford their choice, and society has to afford it too. Healthy food was twice as expensive as processed food in 2002, and its cost increased by the equivalent of US $0.22 per year over the next 10 years, compared with processed food, which increased by the equivalent of US $0.09 per year [160].

#### 9.1.4. *Non-Anarchy*

The medical costs of chronic metabolic disease due to sugar consumption will cause a doubling of Medicare costs in the next decade [161], bankrupting health care systems around the world [162,163], and the NHS is under an ever-tighter squeeze, resulting in lengthier waiting times [164]. The argument that your actions cannot harm anyone else ignores the diet-related harm experienced by children who are especially vulnerable to poor diet at critical development stages.

Americans currently consume an average of 19.5 tsp/day of added sugar. The American Heart Association has recommended a reduction in added sugar consumption to 6 tsp/day for women and 9 tsp/day for men, a reduction by ^2^/_3_ to ¾ in amount. Of these 22 tsp, ^1^/_3_ can be found in beverages, and 1/6 in desserts. This means that fully ½ of the added sugar in our diet is in foods that we did not know contained sugar, such as salad dressing, bread, tomato sauce, ketchup, and many other common food items. Thus, even if we removed all the soft drinks and desserts from our diet, we would still be over our “sugar limit”, which has been set so high by the food industry. Thus, “personal responsibility’ alone cannot be expected to confer any relief. Indeed, our food supply has been “adulterated” by the addition of added sugar by the food industry. Furthermore, there are 262 names for sugar, most of which are unknown to the population at large [165]. As the Nutrition Labeling and Education Act of 1990 [166] requires listing food ingredients by mass, the food industry can hide added sugar by using various forms of sugar and thus moving each form further down the label, so that the consumer does not know that the food they are purchasing is laden with added sugar [167]. Furthermore, while each disease within metabolic syndrome can be temporized, there is no pharmacologic “fix” for metabolic syndrome itself. Paracelsus said in 1537: “The dose determines the poison”. Added sugar has an upper limit of 25–37.5 gm/day for adults and 12 gm/day for children; and we have been placed over our limit by the food industry.

The reduction of added sugar from the American diet must become the top priority to reverse the prevalence and severity of NCDs. Prevention strategies of necessity must occur through public health interventions to alter the food environment. But how? Food is a personal choice, most consider sugar as just “empty” calories, and if individuals want to consume their discretionary calories as sugar, why should they not be allowed to do so? Yet, tobacco and alcohol similarly pose significant societal threats due to their abuse, toxicity, ubiquity, and externalities (negative impact on society) [155,168], and they are regulated [169].

### 9.2. Is Added Sugar ‘Food’?

The food industry will debate any argument for regulating added sugar with two talking points. First, they will point out that sugar is a primary component of fruit, and fruit has been shown to be preventive against NCDs [170]. In contradistinction, fruit juice has been shown to be correlated with these same diseases [76,171]. The reason is that the fiber prevents intestinal absorption, thus reducing the systemic burden of the sugar in whole fruit [172]. Second, the industry argues that dietary sugar is on the FDA’s Generally Recognized as Safe (GRAS) list, which gives the food industry license to use any amount of sugar in any foods they wish. Fructose was grandfathered into the first GRAS list in 1958, as it was “natural” and had been used for generations without any obvious ill effects—although sugar was known to be associated with gout as early as the 17th century [173], and known to raise serum uric acid levels (the mechanism of gout) in 1967 [174]. It should be noted that inclusion on the GRAS list prior to 1 January 1958 was through either scientific procedures or experience based on common use in food (requiring a substantial history of consumption for food use by a significant number of consumers) and thought there is reasonable certainty that the substance is not harmful under the intended conditions of use (Food, Drugs, and Cosmetics Act (FDCA) 321(s), 21 CFR 170.30(c), 170.3(f)). However, in 1958 our consumption of added sugar averaged 2 ounces per day, and currently it averages 6.5 ounces per day. Thus, GRAS determinations in 1958 do not hold for today’s food supply. The issue of GRAS outliving its intentions is seen for trans-fats and salt; both used by the processed food industry, both proven to be detrimental at doses above what were thought to be safe, and now both under scrutiny by the FDA (although not removed from the GRAS list).

*Trans*-fats used to be “food”, but subsequent research showed they cause heart disease and other metabolic diseases. Nitrates used to be “food”, yet research showed they cause colon cancer. Both were eventually removed from the GRAS list, and are now regulated as food additives. Ethanol has always been a food additive, and caffeine dosage above 0.02% (in cola drinks) is similarly regulated.

The question is, does added sugar legally qualify as food? It depends on how you define the word “food”. The Food, Drug, and Cosmetics Act (FDCA, 1938) 321.201(f) defines the term “food” as: *(1) articles used for food or drink for man or other animals, (2) chewing gum, and (3) articles used for components of any such article*. The first rule of vocabulary is that you are not allowed to use the word in the definition. The Merriam-Webster Dictionary defines “food” as: *a material consisting essentially of protein, carbohydrate, and fat used in the body of an organism to sustain growth, repair, and vital processes and to furnish energy*. Fructose supplies energy, so that should make it a food. Or does it? Ethanol supplies energy (7 kcal/gm), but it is clearly not a food. There is no biochemical reaction in any eukaryote that requires it. When consumed chronically and in high dose, ethanol is toxic, unrelated to its calories or effects on weight. Not everyone who is exposed becomes addicted, but enough do to warrant public health intervention [175]. Clearly, ethanol is NOT a food, it is a food additive. Similarly, *added sugar is a food additive*—like ethanol, it is not essential for life, it is toxic in chronically high dosage, and a good percentage of the population is addicted. Indeed, the petitioning for removal of fructose from the GRAS list is being currently being entertained by public health non-governmental organizations (NGOs).

## 10. Possible Societal Interventions

In the last 30 years, there have been four global cultural tectonic shifts in behavior to ameliorate four public health problems: (a) smoking in public places; (b) drunk driving; (c) bicycle helmets and seat belts; (d) condoms in public bathrooms. In each case, public education was necessary but not sufficient, and some form of regulatory policy also had to be enacted to insure compliance. There are many lessons from alcohol and tobacco control policies that can be brought to bear on sugar and ultraprocessed food.

### 10.1. Public Education

One the most important things we have learned from tobacco and alcohol policy research is that public education, despite being the most popular and a necessary component of prevention, does not work alone [168,176]. Evidence from the U.S. suggests that government labels warning consumers about the health effects of excessive drinking have no effect on alcohol consumption, but might have had some limited effect on risky drinking patterns, such as drunk driving [177]. The most popular approaches–school-based health education, public information campaigns, product labeling, and government guidelines—do not work in isolation [178,179]. It should be noted that education *alone* has not solved any substance of abuse. Nonetheless, education softens the playing field, so that societal policy interventions can become acceptable and take hold.

We must take a look as to what works to reduce the consumption of addictive substances. Research on alcohol policy demonstrates that regulatory controls on the pricing, marketing, and distribution of alcohol are highly effective worldwide in reducing the negative impacts of alcohol consumption [10,168,176]. This strategy has also been effective with tobacco [180]. There are three ways to reduce availability: pricing strategies (e.g., taxation), restriction of access (e.g., blue laws), and interdiction (e.g., banning). No one thinks interdiction is a good idea—alcohol prohibition was tried, and was singularly unsuccessful.

### 10.2. Pricing Strategies-Taxation

Society accepts taxation because taxes affect only those who use those products. While tobacco and ethanol are significant burdens to society, sugar is by far and away the most expensive burden. The question is, what is the real goal? Making money for the state, or reduction in consumption? Because if you reduce consumption, you limit revenue generation. For a tax on a hedonic substance to work, it has to hurt. Most soda taxes are 10%, but an Oxford group modeled that a soda tax would have to be at least 20% to reduce general consumption [181].

The good news is that due to the emergence of the science around sugar and the inability of education to stop the diabetes pandemic, six American cities and 28 countries globally have enacted sugar taxes, and others are considering some form of legislation [182].

### 10.3. Pricing Strategies-Subsidies

Agricultural subsidies are payments and other kinds of support extended by the U.S. federal government to certain farmers and agribusinesses. They are a holdover from the original Farm Bill of 1933, when it was necessary to provide cheap food to a destitute population across the country. In the U.S., currently seven states are awarded 45% of the subsidies: Texas 9.6%; Iowa 8.4%; Illinois 6.9%; Minnesota 5.8%; Nebraska 5.7%; Kansas 5.5%; and North Dakota 5.3% [183]; and these are the states that that are the largest producers of corn, soybeans, wheat, and rice—the basics for ultraprocessed food production. There is no economist on the planet who believes in food subsidies, because they distort the market. They make available the wrong stuff while making the right stuff harder to afford. As long as commodities are cheap, real food will stay out of reach for much of the population.

What would happen if subsidies ended? The Giannini group at UC Berkeley modeled what food would actually cost; and the only two items that would increase in price are sugar and corn [184], which is just what we would want to happen. Not surprisingly, these are two of the major industries fighting to maintain the status quo. Still, people will argue, the overall price of food will go up. Well maybe it should, The U.S. spends the least percentage of GDP on food of all nations at 7%—that is because all the food is commodity crop-based and processed. The next two are the UK at 9% and Australia at 11%, the three fattest nations [185].

### 10.4. Restriction of Access-Workplace Bans

The workplace presents an educational moment and venue. At UCSF, all sugared beverage sales—soda and flavored coffee drinks—were banned from sale in the cafeterias, vanished from patients’ meal trays, and disappeared from the menus of any vendors bringing food onto campus. We studied a subgroup of 214 employees who regularly drank sugared beverages before and one year after the ban was put in place [159]. They reported a daily intake of 35 ounces at baseline and 18 ounces at follow-up—a 17-ounce decrease, a cut by almost half. In addition, waist circumference reduced by 2.1 cm. Reductions in sugared beverage intake correlated with improvements in waist circumference, insulin sensitivity, and a pattern of reduction in blood lipids. Some employers may face challenges in implementing a workplace culture where SSB sales bans are perceived as paternalistic. Nevertheless, this proves the *Iron Law* does indeed work.

### 10.5. Restriction of Access-Stipends

The U.K. provides people with a monthly stipend—which can only be exchanged for real food [186]. This allows each person to use their stipends to vote on local food policy, and in so doing, promote local farmers and organic practices. 

### 10.6. Combination Strategies-Differential Subsidization

Differential subsidization combines the “carrot and stick” approach—the inducement with the punishment [187]. Differential subsidization was employed in 1977 in the Nordic countries, including Sweden, Denmark and Norway, to curb the increasing number of alcoholics in their respective countries. The three countries collectively adopted two pieces of legislation: first, they nationalized the liquor stores resulting in the same products sold at the same amount everywhere; second, they taxed high-alcohol spirits, and then used the money from the tax to subsidize low-alcohol beer. In doing so, they were able to nudge the public away from hard spirits and toward the low-alcohol beer, thus, reducing alcohol consumption. In the process, hospitalizations decreased, car accidents reduced, cirrhosis of the liver declined, and economic productivity improved [188].

This could easily be used to cut sugared beverage consumption—tax soda, and use the revenue generated from the tax to subsidize water. The beverage makers will not care, because they are also selling the water. It is just a straight up exchange, nudging people to a healthier option with a zero-sum scheme. In so doing, you can “nudge” people into doing the right thing, and they won’t complain—and most of the time, they will not even notice they have been nudged.

## 11. Conclusions

When it comes to public health, personal intervention (read: rehab) must be balanced with societal intervention (read: laws). For tobacco, alcohol, opioids, cholera, HIV, lead, pollution, and venereal disease, invoking “personal responsibility” and railing against the “nanny state” was ultimately unsuccessful, and both forms of intervention were ultimately deemed necessary. For added sugar and NCDs, we currently have nothing. The argument for societal intervention in NCDs has been lacking because the food industry has convinced the public that “a calorie is a calorie”, that sugar are just “empty calories”, and that “personal responsibility” is the answer. While public educational efforts are necessary to warn about the hazards of chronic excessive sugar consumption, they will not be sufficient, as has been seen for every other hedonic substance.

Added sugar, like tobacco, alcohol, cocaine, and opioids, meets public health criteria for societal intervention, i.e., regulation. The roadmap to successful intervention is complex, but we have templates based on how tobacco and alcohol regulation were enacted. As with tobacco and alcohol, the *Iron Law of Public Health* is in force, which states that reduction in availability results in reduction in consumption, which results in reduction in health harms [10]. Policies that target availability, affordability or acceptability (e.g., the Mexico sugar tax) are effective in curbing sugar consumption [111]. But similar to what occurred with the tobacco industry (e.g., *Merchants of Doubt*), the sugar industry, their legislative partners, and their political allies have utilized numerous instruments to deflect culpability and derail policy changes. Some involve influencing science, some involve influencing public opinion, and yet others influence legislatures directly [189]. These activities must be understood and countered before any specific and meaningful policy measures can be proffered.

In this article, I have provided evidence that: (1) sugar is addictive and toxic unrelated to calories; (2) sugar reduction confers health and societal benefits; (3) added sugar, and by inference ultraprocessed food, meets criteria for regulation; (4) sugar reduction is not only possible but required to save health and healthcare, and (5) societal interventions to reduce consumption of processed foods containing added sugar are achievable and necessary. Those interventions (administrative, legislative, judicial) will likely be geographically, politically, and culturally specific; and certain policy interventions will not work in certain venues.

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
