# Peer review of "Ultraprocessed Food: Addictive, Toxic, and Ready for Regulation"

_nutrients, 2020, doi:10.3390/nu12113401_

Round 1
Reviewer 1 Report
Thank you for the opportunity to review “Ultraprocessed food: addictive, toxic, and ready for regulation.” This was a very interesting and compelling commentary addressing the health risks and consequences associated with food additives, particularly processed sugar. The author presents a focused review highlighting the association between consumption of fructose and other sugars with non-communicable diseases responsible for significant morbidity, mortality, and significant economic cost. The information is clearly relevant and would be of interest to readers.
A primary concern regarding this manuscript is that it was submitted as an “Article” (i.e., manuscript describing original research), yet it appears to be a non-systematic review/commentary. Related to this, the use of some colloquial language and other aspects of the writing style (i.e., vocabulary that seems to vilify the food industry—rather than simply presenting the facts) give the reader an impression of bias that is typically avoided in objective scientific writing. This is not to take away from the strong case made by the author—more so to suggest that the case may be strengthened by reviewing the paper for these instances and editing them to present the facts more objectively.
A bit of reorganization may improve the paper, as the main thesis is not immediately clear and the paper jumps between several related topics before finally introducing the 4 conditions to consider regulation on page 10. Perhaps the introductory paragraph could be extended to introduce the idea that ultraprocessed foods are the culprit (i.e., move the information from lines 558-71 to the introductory section), and also include lines 465-467. Then, the paper can follow the outline suggested in that paragraph (i.e., arguments 1, 2, and 3). That would help to structure the paper and limit repetitiveness.
Please consider moving the first subheading (“Obesity is a “marker” not a cause of NCDs”) AFTER the introductory paragraph(s).
On page 2, it is unclear how one would make the leap from the idea that obesity is a cause of NCDs to the suggestion that “eating is addictive.” Please clarify to help connect the dots for the reader.
On page 3, lines 116-120 are somewhat hard to follow. Please consider breaking this sentence into 2-3 sentences to improve comprehension.
On page 6, the description of food as “addictive” is somewhat confusing due to use of inaccurate terms. For example, the tolerance and withdrawal criteria in DSM-IV and DSM-5 are the same, but there is no “substance dependence” in DSM-5 (it was changed to moderate-severe “substance use disorder”). In addition, the DSM-5 only includes gambling disorder alongside substance use disorders in the section on addictive disorders. It would be worth reviewing the work by A. Gearhardt et al. for discussion of the construct of food addiction. In addition, there is evidence of “withdrawal” symptoms related to food—see the recent systematic review on food addiction by E. Gordon et al.
The Conclusion section, in particular, would benefit from editing to better reflect the more technical style of scientific writing.
Other issues:
- There was no graphical abstract included with the manuscript.
- There were several instances where an acronym was presented without first spelling out the name (e.g., U.N., T2DM, NAFLD, CVD—all on page 1). Please spell out the full terms before using abbreviations, even for these common terms.
- The abbreviation for non-communicable diseases should be NCDs, not NCD’s
- There are a number of typos/grammatical errors (e.g., extra or missing words, lack of subject-verb agreement) throughout the manuscript that should be corrected.
- On page 3, there is a long quotation. It may be preferable to indent this to make it more apparent that it is a quote.
- There are a few areas in the manuscript where the font size changes (e.g., page 4, line 167; page 5, line 230; page 11, line 524…)
- The correct abbreviation is DSM-5 (not DSM-V)
- On page 12, lines 565-566, it would be helpful to list the US dollar and/or euro equivalent, which would be more familiar to most readers than the British pound/pence system
Author Response
Reviewer #1
A primary concern regarding this manuscript is that it was submitted as an “Article” (i.e., manuscript describing original research), yet it appears to be a non-systematic review/commentary. Related to this, the use of some colloquial language and other aspects of the writing style (i.e., vocabulary that seems to vilify the food industry—rather than simply presenting the facts) give the reader an impression of bias that is typically avoided in objective scientific writing. This is not to take away from the strong case made by the author—more so to suggest that the case may be strengthened by reviewing the paper for these instances and editing them to present the facts more objectively.
I have gone through the entire manuscript, and removed any editorial comments that could be construed as biased. The science speaks for itself. As to the type of paper, although it does not contain primary data, I review specific points and mechanisms, and the manuscript is a new synthesis of this information and logistics in a way that has not been performed before and that can newly affect policy. Therefore, I thought that it constituted an article rather than a review. I checked this with Ms. Su, the Assistant Editor of Nutrients upon submission, and she agreed.
A bit of reorganization may improve the paper, as the main thesis is not immediately clear and the paper jumps between several related topics before finally introducing the 4 conditions to consider regulation on page 10. Perhaps the introductory paragraph could be extended to introduce the idea that ultraprocessed foods are the culprit (i.e., move the information from lines 558-71 to the introductory section), and also include lines 465-467. Then, the paper can follow the outline suggested in that paragraph (i.e., arguments 1, 2, and 3). That would help to structure the paper and limit repetitiveness.
I have reworked the front of the manuscript to describe its flow and I present the criteria that are necessary for regulation. I then follow the same architecture throughout.
Please consider moving the first subheading (“Obesity is a “marker” not a cause of NCDs”) AFTER the introductory paragraph(s).
I have changed the order of the front end of the manuscript, so that this concept makes sense in its new flow.
On page 2, it is unclear how one would make the leap from the idea that obesity is a cause of NCDs to the suggestion that “eating is addictive.” Please clarify to help connect the dots for the reader.
I have reworked these sections to make it clear how one follows from the other.
On page 3, lines 116-120 are somewhat hard to follow. Please consider breaking this sentence into 2-3 sentences to improve comprehension.
I have split this sentence into two, and restructured both to make the concept easier to understand.
On page 6, the description of food as “addictive” is somewhat confusing due to use of inaccurate terms. For example, the tolerance and withdrawal criteria in DSM-IV and DSM-5 are the same, but there is no “substance dependence” in DSM-5 (it was changed to moderate-severe “substance use disorder”). In addition, the DSM-5 only includes gambling disorder alongside substance use disorders in the section on addictive disorders. It would be worth reviewing the work by A. Gearhardt et al. for discussion of the construct of food addiction. In addition, there is evidence of “withdrawal” symptoms related to food—see the recent systematic review on food addiction by E. Gordon et al.
I thank the Reviewer for the correction. I am very familiar with this line of research, and very familiar with both Ashley Gearhardt and the entire U. Florida-Gainesville team. I have amended this section and included the Gordon review in the References.
The Conclusion section, in particular, would benefit from editing to better reflect the more technical style of scientific writing.
I have reworked the Conclusions to remove editorial comments, and reflect the new structure of the manuscript.
Other issues:
- There was no graphical abstract included with the manuscript.
I do not think this manuscript lends itself to a graphical abstract. If the editors insist on one, I will try to accommodate.
- There were several instances where an acronym was presented without first spelling out the name (e.g., U.N., T2DM, NAFLD, CVD—all on page 1). Please spell out the full terms before using abbreviations, even for these common terms.
I have gone through the entire manuscript to find all abbreviations, and have made sure the the first mention is completely spelled out and elaborated.
- The abbreviation for non-communicable diseases should be NCDs, not NCD’s
I have changed the abbreviation per the Reviewer’s request.
- There are a number of typos/grammatical errors (e.g., extra or missing words, lack of subject-verb agreement) throughout the manuscript that should be corrected.
I have done my best to find all typos and grammatical errors and correct them.
- On page 3, there is a long quotation. It may be preferable to indent this to make it more apparent that it is a quote.
I had indented it in my original manuscript. The indentation was lost in processing. I have restored the indentation.
- There are a few areas in the manuscript where the font size changes (e.g., page 4, line 167; page 5, line 230; page 11, line 524…)
Again, these occurred in processing. I have fixed whatever typo errors I could find.
- The correct abbreviation is DSM-5 (not DSM-V)
I have made this change per the Reviewer’s request.
- On page 12, lines 565-566, it would be helpful to list the US dollar and/or euro equivalent, which would be more familiar to most readers than the British pound/pence system
Using Currency Converter, I have converted British pence to American cents.

Reviewer 2 Report
Excellent manuscript. The only thing that I have missed are some recommendations on appropriate public health measures to minimize the effects of ultraprocessed products. Regarding the content of the manuscript and the research, in general, simply congratulate the author.
Author Response
Reviewer #2
Excellent manuscript. The only thing that I have missed are some recommendations on appropriate public health measures to minimize the effects of ultraprocessed products. Regarding the content of the manuscript and the research, in general, simply congratulate the author.
I appreciate your support. I have added as section toward the end on rational interventions. This has grown the manuscript somewhat, and I hope the Editors will allow it as it was requested.

Reviewer 3 Report
In this review article, the author provided compelling evidence and the rationale for the need to institute public health interventions to curb the consumption of added sugars worldwide.
The author did an excellent job presenting the information and drawing conclusions.
Just two observations:
- The manuscript would benefit from including a discussion of the findings presented in the manuscript referenced below under the section “3. Added sugar meet public health criteria for regulation”
“Faruque, Samir, et al. "The Dose Makes the Poison: Sugar and Obesity in the United States–a Review." Polish journal of food and nutrition sciences 69.3 (2019): 219.”
- Please review the manuscript for typographical errors.
Author Response
I'm sorry, I had already responded to Reviewer 1 and Reviewer 2. This review came after I had finished with the revisions.

Round 2
Reviewer 1 Report
Thank you for the opportunity to re-review this manuscript. I appreciate the author's attention and response to each reviewer comment. The paper-- which was already very interesting, timely, and persuasive-- was improved by the reorganization. Kudos to the author for an impressive synthesis of the science related to this topic.
This paper will make a valuable contribution to the literature and will hopefully contribute to public dialogue and policy-making. As a side note: the author has convinced this reader of the importance of limiting sugar intake, and I intend to make dietary changes with my family as a result!